# Skin Minerals: Key Roles of Inorganic Elements in Skin Physiological Functions

**DOI:** 10.3390/ijms23116267

**Published:** 2022-06-03

**Authors:** Marek Haftek, Rawad Abdayem, Pascale Guyonnet-Debersac

**Affiliations:** 1CNRS Laboratory of Tissue Biology and Therapeutic Engineering (LBTI), UMR5305 CNRS–University of Lyon1, 69367 Lyon, France; 2L’Oréal Research and Innovation, 94550 Chevilly-Larue, France; rawad.abdayem@rd.loreal.com; 3CARITA Brand, 92300 Levallois-Perret, France; pascale.guyonnet-debersac@loreal.com

**Keywords:** mineral elements, skin physiology, skin functions

## Abstract

As odd as it may seem at first glance, minerals, it is what we are all about…or nearly. Although life on Earth is carbon-based, several other elements present in the planet’s crust are involved in and often indispensable for functioning of living organisms. Many ions are essential, and others show supportive and accessory qualities. They are operative in the skin, supporting specific processes related to the particular situation of this organ at the interface with the environment. Skin bioenergetics, redox balance, epidermal barrier function, and dermal remodeling are amongst crucial activities guided by or taking advantage of mineral elements. Skin regenerative processes and skin ageing can be positively impacted by adequate accessibility, distribution, and balance of inorganic ions.

## 1. Introduction

Although the life on Earth is carbon-based, several other elements present in the planet’s crust are involved in and often indispensable for functioning of the living organisms [1].

Water composes 60 to 70% of the human body. It provides the medium for interaction of organic and non-organic molecules that constitute our organisms. Interaction with water is therefore crucial for several life-sustaining functions e.g., absorption of nutrients, tissue growth and differentiation, or disposal of harmful metabolites. It is thus in ionized form that minerals get access to cells and tissues and become integrated into the living matter. In addition to the four bulk organic elements (oxygen, hydrogen, carbon, and nitrogen) that constitute approximately 95% of the human body weight, seven essential mineral macroelements (calcium, phosphorus, sodium, potassium, magnesium, sulfur, and chlorine) are needed for proper functioning of the organism. Several other elements, called oligoelements, are not less essential but required only in smaller or even trace quantities (Fe, Cu, Zn, Mn, I, Se…). They all have specific biochemical functions in the human body [2]. As it comes to the relative quantities of a given mineral, the elemental composition of human skin and its annexes differs in some regards from that of the whole body, e.g., prevalence of sulfur-containing keratins in the epidermis, hair, and nails or absence of the mineralized tissue composed mainly of calcium and phosphorous (Table 1). From a nutritional point of view, minerals may be subdivided into groups of elements that represent different daily dietary requirements. Concerning oligoelements, the quantities are often too minute to be defined as minimal daily intake levels. However, by definition, their presence is essential for sustaining life processes.

Skin constitutes the dynamic interface between the body and the environment. The surface of these exchanges approximates two square meters and, by weight, skin ranks as the heaviest organ of the body. The vascularized dermis, rich in fibrous elements (collagens and elastin fibers) is the mechanically most consistent part of the skin. It provides a solid support and is the nutritive source to the overlying epidermal tissue deprived of blood vessels [3,4]. The final product of epidermal differentiation, the continuously renewed horny layer (*stratum corneum*, SC), forms a relatively impermeable barrier at the top of the skin. This physical fence is crucial for the organism’s survival in the terrestrial environment [5,6]. Its formation and functions rely on complex interactions between structural, chemical, immune, and sensory elements/factors, and interrelate with the commensal flora at the skin surface [7,8,9]. The deepest part of the skin, hypodermis, is where the body stores energy-rich fatty tissue. This latter skin layer is also a thermal insulant, mechanical buffer, immune regulator, and place of vivid hormonal activity implicated in the management of energy resources [10,11].

The elementary mechanisms governing cell and tissue function employ the same mineral elements in the skin as in the other organs. However, the particular role of the epidermis as the outside-in and inside-out permeability barrier and the protective role of the entire integument permanently exposed to environmental aggressions require specific local arrangements. Calcium and magnesium gradients regulate terminal differentiation of keratinocytes resulting in the constitution of the SC barrier, whereas zinc-, magnesium-, or selenium-depending enzymes are particularly active in remodeling of the extracellular matrix and in quenching of free radicals generated by the ultraviolet radiation and pollution [12].

This review focuses on the involvement of macro- and microelements in the function of human skin.

## 2. Mineral Elements in Skin Homeostasis

Most of the mineral elements present in living tissues occur in ionized form, dissolved in the body liquids and, obviously, do not interact as a dry mineral matter. This is also the case in the skin where they occur in ionized form in its three main components: the epidermis, dermis, and subcutaneous tissue [13,14]. Sulfur-rich so-called “hard keratins” of hair and nails are exceptions, and so are the protein-bound metals involved in transport and enzymatic processing of other molecules, e.g., iron in the heme, selenium in glutathione peroxidases [15], or zinc in matrix metalloproteinases. Distribution of ions in the living parts of the skin is subjected to the local “requirements” for a given element and depends on the systemic (blood) availability of ions and their ability to diffuse through the tissues, e.g., affinity to the carriers and the tissue structural elements such as proteins and lipids. Dependence on pH and redox capacity of the cell and tissue compartments is also an issue [16]. In the skin, elements and nutrients are supplied from the dermal capillaries to the non-vascularized epidermis through the dermal-epidermal junction, as they are carried by the interstitial fluid.

Detection and quantitation of elements depend on the methods used, i.e., their sensibility and accuracy [17,18,19,20,21,22,23,24,25,26]. The outcomes may also vary due to the conditions in which the specimens are studied (in vivo, lyophilized, previously dissociated/dissected tissues). Therefore, the reported results must be examined in the light of sound comparisons taking into account all the variables and in relation with already gained basic knowledge on elements’ role in physiology. Distribution of most abundant minerals in human epidermis is schematically presented in Figure 1.

Another gradient of distribution may apply when the body is subjected to environmental exposure, whether it is fortuitous or on purpose. The topical way of penetration of elements into the skin is subjected to the effectiveness of the natural protective barrier provided by the SC. However, one should consider the fact that many mineral elements also constitute integral building blocks of the living matter, being chemically bound to “organic” molecules such as amino acids, lipids, nucleotides, and carbohydrates.

### 2.1. Cell Formation, Functioning and Division Are Basic Tasks Assisted by Minerals

Elementary biological functions are supported by mineral elements. First of all, cell-delimiting membranes are made of phospholipids, phosphorus-based molecules. Next, genetic information is encoded in the phosphorus-containing nucleic acids (DNA and RNA). Additionally, generation, storage, and consumption of cell energy are adenosine triphosphate (ATP)-based [27]. As can be seen from the above, phosphorus ranks high on the list of life-sustaining elements.

Among the minerals exerting fundamental roles in cell biology, magnesium is of paramount importance, since such processes as high-fidelity DNA replication and continuous DNA repair are dependent on the presence of magnesium ions [28]. Magnesium acts as a cofactor in several proteins involved in these processes. By binding phosphate groups of polynucleotides, Mg^2+^ ions stabilize the double helix and help to electrostatically maintain the compact state of heterochromatin. Cell division as well as the regulation of the cell cycle, proliferation, differentiation, and apoptosis require the presence of magnesium ions originating mainly from intracellular pools, i.e., mitochondria, the nucleus, and the endoplasmic reticulum. At the cellular level, magnesium is one of the most abundant inorganic elements.

Proteins expressing “zinc finger” domains are among the most profuse in eukaryote cells [29,30]. Their diverse functions are most frequently related to the capacity of zinc finger motives to bind nucleic acids. These include DNA recognition, transcriptional activation, RNA packaging, nuclear hormone receptors binding, and regulation of apoptosis. However, protein folding and assembly, and also lipid binding, remain within the field of action of the zinc finger domains. In the skin, at least two zinc finger-containing proteins have been reported to participate in the regulation of terminal differentiation of keratinocytes and epidermal barrier formation, i.e., ZFP36 [31] and Krüppel-like transcription factor KLF4 [32].

Zinc ion homeostasis is regulated at the cellular level by numerous dedicated transporter proteins that control Zn^2+^ efflux and uptake. ZIP7 and ZIP13 transporters have been shown to be involved in the formation of dermis and proper fibroblast function, ZIP2 and ZIP4 are essential for keratinocyte differentiation and proliferation, ZIP13 and ZIP14 for adipocyte biology, whereas ZIP10 is associated with the keratinocyte progenitors in the outer root sheath of hair follicles. Zinc deficiency is responsible for acquired acrodermatitis enteropathica, necrolytic migratory erythema, and pellagra. It also has profound impact on alopecia and delayed wound healing [33,34]. Metallothioneins are ubiquitous cytoplasmic storage proteins capable of biding zinc, copper, and cadmium. Buffering the excess of Zn and releasing it in the condition of zinc deficiency, they are important factors maintaining homeostasis of this element [35].

Sulfur is one of the most chemically versatile elements, which makes it essential to the life and growth of all organisms. It plays a crucial role in the regulation of diverse biological processes in the human body. Organic compounds containing a carbon–sulfur bond can form a variety of molecular arrangements and exhibit different biological activities [36]. Sulfur amino acids participate in the synthesis of essential biomolecules such as antioxidants, vitamins, and co-factors such as thiamine, lipoic acid, biotin, or coenzyme A [37]. Sulfur-containing L-cysteine and homocysteine are principal organic sources of hydrogen sulfide H_2_S, a gaseous messenger active in tissue biology, along with nitric oxide (NO) and carbon monoxide (CO) [38,39]. Synthetized by ubiquitously expressed enzymes, mainly cysteine aminotransferase/3-mercaptopyruvate sulfotransferase pathway, cystathionine ß-synthase, and cystathionine γ-lyase, endogenous H_2_S is highly active in endothelium-dependent vasorelaxation of dermal blood vessels [40] and in antioxidant processes [41]. Both biological effects are of great interest for skin microcirculation and detoxification.

### 2.2. Cell Metabolism and Bioenergetics

The term bioenergetics encompasses all vital functions that require energy production and its expenditure; in other words, it describes metabolism events that relate specifically to energy homeostasis. The main powerhouse of mammalian cells are mitochondria that perform redox reactions and use the generated energy to phosphorylate adenosine diphosphate (ADP) to adenosine triphosphate (ATP), i.e., to increase of the phosphorylation potential. The process is closely related to the simultaneous generation of free radicals that require subsequent neutralization (as described further).

#### 2.2.1. Cellular Energy Production (ATP)

Vital bioenergetic processes, such as cellular respiration, are at the root of functioning of living organisms, and mineral elements play a crucial role therein. Specifically, respiratory chain activity in mammalian cells is supported by mitochondria where iron plays a key role as cofactor of several enzymes. During the reaction, glucose is oxidized to carbon dioxide and water while the released energy is captured by the energy-carrying ATP. The other two ways to generate ATP are acetyl-coenzyme A-driven tricarboxylic acid cycle (Krebs cycle) and ATP synthase-mediated oxidative phosphorylation of ADP. Inorganic phosphorus required for this reaction is polymerized and stored in cells [42,43], and this process appears to be reciprocally regulated by cell energy metabolism [44,45]. Numerous steps in the catabolic transformation of nutrients (proteins, carbohydrates, lipids) require intervention of magnesium-dependent enzymes (Figure 2).

Additionally, all reactions involving ATP require the presence of magnesium ions that minimize electrostatic repulsion between negative charges of ATP and ADP (ATP-Mg^2+^ and ADP-Mg^2+^ complex) [46,47,48].

#### 2.2.2. ROS Balance

Natural by-products of cellular respiration are free radicals with the superoxide anion radical (O^−2^) in the first place. This highly reactive oxygen species (ROS) would be harmful to cells and must be eliminated by partitioning of the radical into less damaging molecules, i.e., ordinary molecular oxygen (O^2^) and hydrogen peroxide (H_2_O_2_). The task is achieved by metal-associated enzymes, superoxide dismutases (SOD). SOD1 is a cytoplasmic copper- and zinc-containing enzyme, whereas mitochondrial SOD2 is manganese-dependent. The mitochondrial superoxide dismutase 2 (SOD2, MnSOD) is a subject of particular interest, as it is located in the mitochondrial matrix where it represents the first line of antioxidant defense against superoxide anions produced as byproducts of oxidative phosphorylation. In human senescent skin, fibroblasts, which develop a growth arrest, morphological and functional changes, and increased ROS concentrations, have been demonstrated in vitro and in vivo with an adaptive upregulation of the SOD2 on mRNA and protein levels, providing evidence for a common response phenotype of cellular senescence [49]. Extracellular copper and zinc-bearing EC-SOD (SOD1) is binding to heparane sulfate proteoglycans at the cell surfaces and in intercellular spaces, being therefore ideally located to prevent ROS-induced damage [50]. Further steps of ROS reduction involve iron-based catalase, which catalyzes the decomposition of hydrogen peroxide to water and oxygen [51], and several peroxidases scavenging various free radicals.

Besides aerobic cellular metabolism that generates ROS, these highly reactive chemical molecules may also be formed in cells by a variety of exogeneous agents such as pollutants, drugs, xenobiotics, or radiation. This latter source of ROS is particularly relevant for the skin, which remains constantly exposed to environmental insults. Two seleno-dependent glutathione peroxidases, glutathione peroxidase and phospholipid hydroperoxide glutathione peroxidase, are enzymes reducing different hydroperoxides to the corresponding alcohols. The latter one is particularly important for detoxification of cell membranes [52]. This selenium-dependent system destroys H_2_O_2_, lipid hydroperoxides, and other organic peroxides.

Free radical reactions are part of normal human metabolism. They are involved in regulation of protein phosphorylation, intracellular signal transduction, regulation of the cytosolic calcium concentration, regulation of gene expression, intracellular killing of bacteria by neutrophils and macrophages, and activation of certain transcription factors, such as nuclear factor-kappa B (NF-κB) and the activator protein 1 (AP-1) family factors [53,54]. Mitochondrial ROS generation plays a regulatory role in the differentiation of various epithelial structures and cell lineages by activating the Notch and β-catenin signaling cascades [55].

Calcium may well be critically involved in this activation pathway since mitochondrial respiration requires Ca^2+^ uptake, whereas extracellular Ca^2+^ levels and Ca^2+^ uptake by mitochondria regulate keratinocyte differentiation in vitro [56]. In agreement with this point of view, a Ca^2+^ gradient has been observed in the epidermis in vivo [57]. Moderate levels or bursts of ROS are thus beneficial to cell and tissue functions in normal conditions. However, exaggerated, prolonged, and cumulative ROS production constitutes a threat and is recognized as one of the major factors that accelerate ageing [58] and promote cancerogenesis [59]. Mineral elements involved in the activity of antioxidant enzymes (Cu, Zn, Mn) influence the redox balance in an essential way, including situations when metal ions such as iron, cobalt, chromium, or copper constitute the source of reactive radicals [60]. Indeed, zinc has been shown to compete with redox-active metals, such as iron and copper, for binding to cell membranes. Unbound elements remain thus available for metal-binding circulating proteins for metal sequestration and storage [61]. Recent experiments with probiotic isolates permitted bioaccumulation of selenium and zinc in the blood of mice fed with the selenium- and zinc-enriched lactobacillus strain. The fact that the animals also showed increased antioxidant capacities indicates new perspectives in mineral-induced modulation of the redox status [62].

### 2.3. Epidermal Differentiation and Barrier Function

The epidermis is a multilayered epithelial tissue in constant turnover. Its principal function is to provide, at the skin surface, a relatively water-impermeable layer of stacked fully keratinized cells embedded in a hydrophobic, lipid-rich extracellular matrix [63,64]. This final product of epidermal terminal differentiation, the SC barrier, undergoes perpetual recycling. Superficial corneocytes desquame while the constant thickness of the layer is maintained through a compensative cornification of the underlying granular layer keratinocytes. Living epidermal cells form a solid, highly interconnected tissue [65] (in press). A dense network of gap junction channels (connexons) results in electrical and metabolic coupling of keratinocyte cytosols within the tissue. As cells are capable of regulating the opening of the connexons, fluxes of ions, such as calcium, sodium or potassium, may be instantly controlled and contribute to the concerted behavior of gradually differentiating keratinocytes within the epidermal living layers. The junctions themselves are calcium-dependent, as their constitutive proteins become fully functional only in the presence of sufficient quantities of Ca^2+^ [66]. Thus, keratinocyte cohesion relaying on classical and desmosomal cadherins, intercellular communication through gap junctions, and compartmentalization of uppermost epidermal layers by tight junctions all require the presence of calcium ions.

It has been established that extracellular calcium in the inter-keratinocyte spaces of epidermal living layers forms a positive distribution gradient spanning from the basal to the granular layer [67]. This increase in Ca^2+^ ions is concomitant with the growing expression of calcium-dependent cadherins, transmembrane proteins of mechanical cell-cell junctions, observed both in vivo and in cell culture submitted to low or high calcium conditions. This confirms the essential role of calcium ions, both extra- and intra-cellular, in the formation of mature desmosomes and, thus, epithelial stratification [66,68,69]. In parallel, a potassium gradient [70] and a magnesium gradient [71] have also been observed.

The epidermal calcium gradient is attenuated or highly disturbed in aged or inflamed skin. Modification of the Ca^2+^/Mg^2+^ ratio in favor of the latter mineral proves beneficial for recovery of the epidermal homeostasis, resulting in an optimal reparation of skin barrier, and may have impact on inflammatory skin diseases [72,73,74]. Acute abrogation of the extracellular calcium gradient by removal of the overlying SC barrier results in an instantaneous mobilization and excretion of the intracellular pool of lipids, aimed at the rapid restoration of tissue impermeability [75]. Many biochemical mechanisms, including those involved in keratinocyte differentiation, are regulated by calcium. Specifically, high concentration of Ca^2+^ induces expression of PAD1 both at the protein and RNA levels in cultured normal human keratinocytes [76]. Peptidylarginine deiminases (PADs) are enzymes involved in protein deamination (or citrullination), one of the crucial steps in processing of filaggrin and keratin during keratinocyte cornification [77]. Ca^2+^-dependent transglutaminase-1 is responsible for the cross-linking of cornified envelope precursors in keratinocytes undergoing terminal differentiation. In this way, it proves essential for formation of the functional SC barrier [78]. On the other hand, proteolytic action of enzymes, involved in SC desquamation is also regulated by the presence of calcium ions. The desquamative action of kallikreins, via degradation of inter-corneocyte junctions, is largely enhanced in the absence of extracellular calcium [79]. These considerations are of importance because a rise in the extracellular Ca^2+^ is able to increase the intracellular calcium pool [80], and the above-mentioned various calcium-dependent mechanisms associated with epidermal differentiation appear to be orchestrated simultaneously [81,82,83].

Another mechanism of control of the intracellular mineral ion distribution relays on cell and mitochondrial membrane ion pumps. A whole host of membrane proteins that mediate conduction of ions or molecules into and out of the cell exist and are expressed in various cell types. Many are ubiquitous, some show skin enrichment profiles, but none are specific to skin cells. Fine regulation of ion levels in the cytosol and its different compartments is crucial for correct cell functioning, as illustrated by dyskeratosis and acantholysis occurring in hereditary diseases due to mutations in sarco/endoplasmic reticulum Ca^2+^ ATPases’ (SERCA) genes coding for calcium transporters [84,85].

Last but not least, zinc, copper, and manganese differentially modulate expression of integrin subunits on cultured keratinocytes promoting experimental epithelial wound healing, an observation tending to explain the beneficial influence of these mineral ions in dermatological practice [86]. Indeed, Zn, Cu, and Mn gluconates have been shown to enhance keratinocyte migration related to an increase in the expression and/or function of integrin receptors, mainly those composed of alpha 2, alpha 3, and beta 1 subunits. Interestingly, signaling through integrins bearing beta 1 subunit is characteristic of the epidermal stem cells with high proliferative potential [87]. Taken together, these observations clearly show a functional effect of oligoelements on keratinocytes’ regenerative capacity.

### 2.4. Dermal Remodeling

Just like the epidermis, the dermal component of the skin undergoes constant remodeling. However, changes in the extracellular matrix of dermis occur on a longer time scale than that observed in the epithelial tissue. This is due to a lower cellularity and a further distance from the skin surface, what equals to a weaker exposure to the deleterious influences from the environment. Most of the changes concern the papillary dermis situated directly beneath the epidermal basement membrane. This zone is rich in elastin fibers and fibroblasts and contains abundant glycosaminoglycans and entropically (dis)oriented bundles of collagen fibers [88]. It is traversed by blood and lymph capillaries, terminal nerve endings, sweat tracts, and pilo-sebaceous complexes. Roots of the hair shafts and secretory parts of the sweat glands are situated deeper, at the limit of or within the reticular dermis, where fibroblasts are less frequent and tightly packed, dense collagen fiber bundles predominate. The synthesis of mature elastin and collagen can be controlled by the availability of copper [89]. Indeed, Cu is a cofactor indispensable to the activity of lysyl oxidase, the enzyme involved in collagen and elastin cross-linking. Phosphore-containing nucleotide adenosine monophosphate (AMP) is part of the structure of coenzyme A, capable of stimulating collagen production by fibroblasts [90]. Zinc is probably also involved in the formation of dermis, as suggested by the reduced thickness of this structure in mice with knocked out Zn transporter gene ZIP7 and the absence of differentiation to fibroblasts of human mesenchymal stem cells with ZIP7 knockdown [91].

Collagen is one of the most abundant structural proteins of the intercellular matrix and a ubiquitous element of the connective tissue. Hydroxyproline and hydroxylysine, both of which are unique to collagen, become glycosylated at specific residues by galactosyl and glucosyl transferases, which require Mn^2+^ as an essential cofactor [92]. The glycosylation step precedes formation of the pro-collagen triple helix and secretion of the molecule into the extracellular space. It may be speculated that sugar moieties help protect collagen from precocious proteolytic degradation and contribute to stabilization of collagen fibers, in addition to molecular cross-linking by lysyl oxidase, which requires copper [93]. Manganese and copper play, thus, vital roles in collagen biosynthesis.

As skin tension changes due to the body movements and variations of the volume of subcutaneous tissue, the collagen fibers must be continuously rearranged along the evolving tension lines. Matrix metalloproteinases (MMPs) secreted primarily by dermal fibroblasts are necessary for removal of denatured or obsolete fiber arrangements that must be replaced with newly synthesized structures. These enzymes belong to the vast family of calcium-dependent zinc-containing endopeptidases [94]. The four main MMP groups present in the skin are the collagenases, the gelatinases, the stromelysins, and the membrane-type MMPs (MT-MMPs). Collagenases are capable of degrading triple-helical fibrillar collagens. Their main skin representative is MMP 1. MMPs 2 and 9 are gelatinases, widely expressed in human skin. The main substrates of the latter are gelatin and type IV collagen constituting the lamina densa of the dermal-epidermal junction. Involved in cell migration, differentiation, inflammation-associated apoptosis, and angiogenesis, MMPs have multiple roles in skin physiology, but also participate in pathological conditions [95].

### 2.5. Skin Regenerative Processes and Wound Healing

Skin regeneration processes, such as epidermal renewal and extracellular dermal matrix biosynthesis, require high amounts of energy [96]. Therefore, mineral-dependent generation of energy resources is of particular importance during wound healing and in fight against sagging metabolic processes related to skin ageing.

In keratinocytes and fibroblasts, inorganic polyphosphates are localized to mitochondria, nuclei, cytoplasm, and cell membranes, and play important role in cell survival and motility [97]. Adequately, in vitro cell culture studies and animal wounded skin models demonstrated beneficial results of polyphosphates on wound healing [98]. Involvement of inorganic polyphosphates in the blood clotting cascade evidently contributes to the wound hemostasis in vivo, an important step in the wound closure. Other processes that inorganic polyphosphates are implicated in are signal transduction, Ca^2+^-signaling, and regulation of the mitochondrial membrane potential [99]. Most recently, the biotherapeutic potential of the phosphate polymers has been reported in the fight against SARS CoV-2, as these physiological inorganic molecules appear to be capable of improving the epithelial integrity as well as the mucus barrier [100].

As mentioned, calcium and magnesium balance, as well as the zinc ions, have fundamental influence on keratinocyte proliferation and differentiation [31,66,71,101]. Therefore, for epidermis, sensing changes in the concentration of these minerals in the tissue is of crucial importance for implementation of re-epithelialization after wounding. Ca^2+^ sensing receptor expressed on keratinocytes conveys information on the extracellular calcium levels and promotes E cadherin-mediated cell adhesion, differentiation, and survival [69].

Zinc was observed to counter calcium-mediated apoptosis [102,103]. Its antioxidant properties are attributed to Zn/Cu superoxide dismutase 1 and the protection of ROS-sensitive sulfhydryl groups on metallothionein, a small cysteine-rich antioxidant protein [104,105]. In vitro, enhanced keratinocyte migration leading to accelerated epidermal wound closure has been observed with zinc, copper, and manganese, apparently via modulation of keratinocyte integrins [86].

On the dermal side, de novo synthesis and reorganization of fibers is supported by zinc, as suggested by modifications in fibroblast biology following invalidation of expression of Zn transporter proteins ZIP7 and ZIP13 [106].

Dermal and epidermal processes engaged for skin regeneration or during wound healing are finely coordinated and complementary [107]. Skin regeneration is a highly energy-consuming activity requiring, besides the “building blocks”, substantial amounts of magnesium that, as already mentioned, mediates and accompanies several metabolic cascades. Mg, Ca, Zn, and other minerals are engaged both in the processes of dermal healing and restoration of the epithelial permeability barrier, thus largely contributing to the successful outcome.

### 2.6. Mineral-Dependent Skin Functions Face to Ageing

Bioenergetic functions decline with age. As is common knowledge, human ageing is characterized by a gradual reduction in the ability to coordinate cellular energy expenditure and storage (crucial to maintain energy homeostasis), and by a gradual decrease in the ability to mount a successful response to stress. Human ageing is accompanied by a decrease in resting metabolic rate, the largest component of total energy expenditure, which significantly affects disability and morbidity among the elderly. Several observations link the ageing process and the age-associated disease with mitochondrial dysfunction [108]. As measured in skeletal muscle cells, we lose about 5% of mitochondrial capacity of energy production over every 10 years of life [109].

According to the free radical theory of ageing, oxidative damage initiated by reactive oxygen species is a major contributor to the functional decline that is characteristic of ageing [110,111].

Generation and accumulation of ROS in mitochondria has been proposed as a likely mechanism involved in tissue ageing and degenerative transformation. Most theories of ageing focus on the idea that cumulative oxidative stress leads to mitochondrial mutations, mitochondrial dysfunctions, and oxidative damage to proteins, membrane lipids, and nucleic acids. Oxidative stress is a condition in which the balance between production of ROS and level of antioxidant defenses is significantly modified and results in damage to cells by the excess of ROS. Cellular levels of ROS are controlled by the antioxidant system, which involves antioxidant enzymes and small-antioxidant molecules, e.g., superoxide dismutases (SODs), glutathione peroxidases (GPx), catalase (CAT), glutathione tripeptide (GSH), and vitamin E. In case of reduced production of or access to these antioxidants, free radicals begin to accumulate oxidative damage and the genesis of the ageing process ensues [108].

Skin is directly exposed to physical and chemical stress from the environment, and the resulting damage accumulates over the years, whereas the defense mechanisms tend to decrease with age [112,113]. The SOD-based antioxidant enzymatic system is located in mitochondria themselves and in the cytoplasm. These enzymes, containing, respectively, Mn and Zn-Cu atoms are able to readily transform O^−2^ free radicals to H_2_O_2_. Next, hydrogen peroxide is decomposed to water and molecular oxygen by cell catalase and Se-dependent glutathione peroxidase, present in both intracellular compartments. Mn-dependent SOD2 is particularly important in quenching superoxide anion byproducts arising during oxidative phosphorylation because of its localization within the mitochondrial matrix. Experimental deficiency for mitochondrial superoxide dismutase in the connective tissue of mice results in accelerated ageing [49]. In this regard, mitochondrial DNA (mtDNA), a circular molecule comprising 16,569 bp in human cells, is particularly vulnerable, due to its close proximity to the mitochondrial electron transport chain (the major intracellular source of ROS), its lack of histones, and a limited repertoire of DNA repair capacity. Mutations of mtDNA are found in mitochondrial diseases but are also frequently detected in aged tissues with high energy demands such as skeletal muscle, heart, and neurons. It has therefore been proposed that mtDNA mutations are causally related to the ageing process. According to this point of view, ageing is caused by the accumulation of mutations and large-scale deletions in mtDNA, partly resulting from the oxidative damage but also arising from the inherent error rate of mtDNA polymerase. Chronic exposure to UV radiation induces deletions of mtDNA in human skin fibroblasts in vitro as well as in vivo, and UV-induced mtDNA mutagenesis is associated with a decline of mitochondrial functions [114].

Nutritional supplementation with selenium and zinc, above the normally required levels, results in only a slight increase in antioxidant activity, partly due to the variable, tissue-dependent availability of these elements [115]. In contrast to selenium, indispensable for the enzymatic activity of GPx, zinc plays only a structural role in SOD and performs its anti-ROS action as a free element by quenching sulfhydryl groups on proteins, thus protecting them from oxidation.

Activation of Zn-dependent metalloproteases is a recurrent phenomenon observed in the course of chronic inflammatory state, typical of aged tissues. When the enzymatic removal of functionally altered elements of the dermal extracellular matrix is not followed by compensatory biosynthesis, the aged skin loses its mechanical properties [116,117]. Long ultraviolet rays (UVA), which can penetrate into the papillary dermis, provoke cumulative long-term damage to its cellular and fibrillary components. UV-induced expression of MMPs adds to extracellular matrix degradation and accelerated skin ageing [117].

A reduced efficacy of the skin barrier function is observed in aged skin. This may be in relation with the inflammatory dermal context, since proinflammatory cytokines are observed to interfere with the barrier permeability [118,119]. In turn, abrogation of the barrier results in disappearance of the epidermal intercellular calcium gradient [75], and vice versa, absence of the physiological gradient impacts normal formation of the SC. Since the activities of most of the epidermal enzymes, including those involved in the regulation of pH, seem to depend on the Ca^2+^ gradient, it has been recently proposed that age-related attenuation of the gradient could contribute to the observed rise of pH at the skin surface in aged subjects [120,121]. Increased pH favors the catabolic action of kallikreins involved in SC desquamation, thus possibly further impeding barrier function [122,123]. In this way, the overall reduction of metabolic activity in the aged skin gets captured in a vicious circle of the calcium gradient collapse and epidermal barrier disruption.

As the epidermal barrier repair capacity declines in the elderly [124], therapeutic manipulation of the Ca^2+^/Mg^2+^ ratio in the epidermis showing compromised barrier function could be suggested as a beneficial means of fighting against skin ageing [73]. Indeed, topical application of 10 mM aqueous solution of magnesium chloride, magnesium sulfate, and magnesium lactate accelerated barrier repair in rodents. Ten mM CaCl_2_ delayed the process, but a mixture of calcium chloride and magnesium chloride still accelerated barrier recovery when the calcium to magnesium molar ratio was lower than 1 [72].

Epidermal calcium gradient is physiologically controlled by the calcium-sensing receptor on keratinocyte membranes. Age-related decrease of the expression of this receptor has been found in mice and humans, resulting in an impaired Ca^2+^ signaling and altered cadherin function [125]. An alternative anti-age approach based on enhancement of keratinocyte sensitivity to calcium could thus be proposed.

Additionally, flattening of the dermal-epidermal junction (DEJ) is observed in the aged skin. This may be both the reason and result of diminished exchanges between these two skin tissues, notably associated with the reduced supply of nutrients and oligoelements. In a study of abdominal human skin, DEJ surface area, as reported to 1 mm^2^ of skin surface, has been shown to be reduced from 2.64 mm^2^ in subjects aged 21–40 years to 1.90 mm^2^ in persons aged 61–80. Such a significant loss of the DEJ convolution appears to contribute to the increased fragility observed in aged skin [126].

Prevention or treatment of epidermal and dermal changes observed during intrinsic and environment-induced ageing could possibly benefit from findings elucidating mechanisms of wound healing: a tissue repair approach implicating mineral elements as well.

## 3. Use of Minerals in Dermatology

Mineral balance is of crucial importance for the fundamental biological functions of the skin. An adequate amount of each and every element is required for proper functioning of the tissues, meaning that deficiency, and sometimes also excess, may cause problems and could be associated with pathologies [14]. As previously mentioned, the trace elements zinc, copper and manganese are used in vivo in some local preparations because of their healing properties. Zinc actually plays a primordial role in skin physiology, in particular it produces a beneficial effect on the healing of leg ulcers by modulating cutaneous inflammation and speeding up the re-epithelialization process. It stimulates the proliferation of epidermal cells (keratinocytes and fibroblasts) in wounds and increases collagen synthesis. In fact, zinc deficiency can often delay healing [86].

Dietary and transcutaneous compensations are possible, provided the implementation of scientifically proven and strictly controlled pharmacological approaches. Proper equilibrium of various elements, in quantities adjusted to the right localization and time, are of critical importance for tissue biology [127]. For example, low micromolar concentrations of hydrogen sulfide support generation of cell energy in mitochondria [128]. However, as generally known, higher concentrations of H_2_S (3–30 μM) suppress mitochondrial function by inhibiting cytochrome-c oxidase (CcOX; complex IV), which makes exogenous gas highly toxic for living organisms [129].

The principal route of mineral ion intake to the skin is through the blood supply and originates from alimentation. The passage of topically applied exogenous minerals into the skin occurs against the natural gradients and requires specific conditions, first of which are ionic dissociation in water and full hydration of the skin. In fact, opening of the paracellular route for hydrophilic molecules in hydrated SC, via “water channels” arising within the hydrophobic inter-corneocyte lipid matrix, appears to be the prerequisite for passive transcutaneous penetration of ions [130]. Hair follicles represent privileged sites of such absorption due to the highly reduced thickness of the SC barrier within the infundibulum and the micro storage function resulting in a prolonged contact with the topically applied solutes containing nanoparticles or mineral elements such as magnesium [131,132].

Bathing in mineral-rich spring water has been advised for treatment of various dermatological affections since ancient times [133,134]. Notably, inflammatory skin diseases such as psoriasis and atopic dermatitis were reported to benefit from such “cures” [135]. The mineral-rich solutes used contain principally magnesium and/or sulfur salts [74].

### 3.1. Immunomodulatory Effect

The anti-inflammatory effect of often incompletely defined mineral components of spring waters is apparently obtained through suppression/modulation of immune cell responses mediated, for example, by effector T lymphocytes [136], antigen-presenting Langerhans cells [18], mast cells [137], or molecular effectors [138,139]. Sulfurous compounds are most frequently incriminated as the most active elements. Anti-inflammatory action of hydrogen sulfide, H_2_S, is due to its interaction with both major inflammatory signaling pathways: NF-κB and nuclear factor-erythroid 2-related factor 2 (Nrf2) [140,141,142]. Not surprisingly, H_2_S has been shown to exert positive influence on cutaneous wound healing, both at its epidermal and dermal component [143,144] and on melanocyte activity [145]. Detoxification of sulfides through oxidation and their conversion to harmless sulfates is efficiently performed by cell enzymes. Non-enzymatic processing of inorganic sulfur contributes relatively little to H_2_S production, however locally delivered sulfide salts, including NaHS and Na_2_S, and other H_2_S pharmacological donors retain their therapeutic potential [146]. In a large-scale study of thermal waters subdivided into groups based on their ionic composition, the anti-inflammatory action of sulfurous molecules has been scientifically validated [147]. Sulfurous bicarbonate and chlorinated sodic waters reduced NO production and/or inducible NO synthase expression, and/or showed scavenging activity in in vitro activated mouse macrophages. However, possible impact of other mineral components of thermal waters studied have not been evaluated, although their presence could have significant impact on diverse cell functions. Indeed, in vitro and/or in vivo studies could demonstrate anti-inflammatory potential of zinc [148], magnesium [149], selenium [150], and copper [151]. Additionally, local binding of copper to non-steroidal anti-inflammatory drugs (NSAIDs) enhances anti-inflammatory potential of the latter [152]. This in situ action of copper ions may reveal interesting when topical Cu supplementation is considered for increasing action of systemic treatment of inflammatory skin disorders.

Organic salts of gold are in current use for systemic treatment of chronic arthritis [153]. Transcutaneous delivery of such compounds could prove of interest in management of inflammatory reactions, notably of small joints underlying skin. However, gold retention in the keratinizing tissues is low during systemic chrysotherapy and mostly concentrates in the dermis [154].

### 3.2. Bactericidal Action

Several mineral elements exert antimicrobial effects. Skin friction with metallic gold was once advised in traditional medicine for small bacterial infections, but it remains unclear whether the observed action was not rather dependent on copper in metal alloys. Indeed, metallic copper and its alloys are well known for germ-inhibiting properties. Solutes containing silver salts, iodine, or potassium permanganate and selenium can be applied to disinfect skin and treat local infections. Wound dressings also take advantage of bactericidal properties of Ag+ or povidone iodine in the management of chronic infected wounds [155,156]. Manganese, iodide, and sulfur present in thermal waters show interesting bactericidal properties as well [157]. Emollients containing copper sulfate are often used as maintenance therapy of easily infected atopic skin. Anti-inflammatory effect of zinc ions from ZnO ointments is due to lowering levels of inducible nitric oxide synthase and NO production, associated with a slight anti-bacterial action [158].

Interestingly, anti-viral (SARS CoV-2) action in vitro of a well-known gold-salt preparation used for oral treatment of refractory arthritis, has been reported recently [159]. The drug, presenting also combined anti-inflammatory and anti-ROS properties, is clearly of interest if it proves efficient in vivo.

### 3.3. Keratolytic Effect

The use of sulfur was once widespread in dermatological disorders such as acne vulgaris, rosacea, seborrheic dermatitis, dandruff, pityriasis versicolor, due to the antibacterial, antifungal, and keratolytic activity of *sulphur praecipitatum*-containing preparations [160,161,162]. Accelerated desquamation of accumulated horny cells is helpful in various dermatoses. It helps to regain skin suppleness and prepares a better access for topical treatments of the lesions. Any xenobiological contaminants (bacteria, yeast, dermatophytes) are also eliminated with the superficial squames.

Nonetheless, scientific evidence of efficacy of topically applied minerals from natural sources is scarce and circumstantial, whereas the underlying mechanisms remain most often unsolved or only speculated upon. Examination of physical influence of composite ceramics and minerals such as tourmaline (boron-silicate) powders on human skin and skin cells in vitro is a good example of evidence-based research yielding yet insufficiently understood mechanism [163,164]. In 2002, experimental work of Yoo et al. has demonstrated that tourmaline and jade radiated far-infrared rays (FIR) and application of 1% tourmaline or jade in emulsion produced elevation of skin temperature by 1 °C [164].

FIR emitted by certain solid minerals under the influence of body heat, has become a hot topic because of its possible biological impact. Even if FIR emission by minerals placed in proximity of the skin can be considered as low compared to FIR heating devices, various observations indicate that it may result in physiological reactions similar to that induced in medical infrared heating settings. Specifically, NO-mediated vasodilation has been suggested as a possible mechanism of positive FIR influence on wound healing [165]. So far, experimental results converge rather towards a possible antioxidant mechanism of action and require further research [166]. However, a direct effect of FIR on dermal fibroblasts, promoting TGF-beta secretion and collagen production, has also been proposed based on wound healing studies in rats [167].

## 4. Conclusions

As seen from this recapitulation of the involvement of various mineral elements in human biological functions, many ions are indispensable for life [2,16]. They are also operative in the skin, supporting specific processes related to the particular situation of this organ at the interface with the environment (Table 2). Skin bioenergetics, redox balance, epidermal barrier function, and dermal remodeling are amongst crucial activities guided by or taking advantage of mineral elements. Skin regenerative processes and skin ageing can be positively impacted by adequate accessibility, distribution, and balance of inorganic ions.

Nutrition is the principal source of elements that make our bodies. Many of the minerals mentioned in this review are found in the underground and sea waters. Their use for systemic and local treatment of skin conditions is recognized since ancient times, e.g., balneotherapy and “cures” with drinking mineral waters [134,168,169]. Although infrequent are well-conducted studies providing scientifically sound conclusions on the exact therapeutic effects of various mineral components on specific skin functions, the potential for such intervention is high. Supplementation with minerals should be science-based and pharmacologically controlled, taking into account particular local aspects in case of their topical delivery to the skin [170]. Clearly, ageing skin and several skin conditions deserve to better benefit from direct contribution of mineral elements delivered to the integuments.

**Table 2 ijms-23-06267-t002:** Functions of main minerals in the skin.

Mineral Element	Functions	References
**Calcium**	The principal constituent of mineralized skeleton, **Ca** is a fine regulator of cell function in general, and of the epidermal terminal differentiation, leading to the constitution of the efficient permeability skin barrier function. Mitochondrial respiration, intercellular junction function, and keratinocyte cornification require the presence of Ca^2+^ ions. Also dermal remodeling by MMPs is calcium dependent.	[28,57,73,76,77,78]
**Magnesium**	**Mg** contributes to the formation of the SC barrier, but its fundamental involvement in the metabolic cascade leading to the production and storage of ATP must be underlined. It is critical for oxidative phosphorylation, transcription and repair of DNA, glycolysis, fatty acid degradation, and protein synthesis. All in all, magnesium is the second most abundant intracellular cation, next to potassium. Over 300 enzymes in the human body have magnesium as a cofactor.	[31,46,47,48,56,72,101]
**Phosphorous**	**P** makes part of many cell and tissue structures and processes ranging from the genetic information coding nucleic acids, through the constitution of cell membranes, to energy-carrying ATP. Inorganic polyphosphates are therefore important for cell survival and motility.	[27,97,98,99,100]
**Sulfur**	A very versatile element, **S** participates in diverse biological processes ranging from the synthesis of essential biomolecules, enzymes, and antioxidants to molecular signaling via hydrogen sulfide. Its anti-inflammatory and keratolytic action is well recognized.	[141,144,162]
**Sodium and Potassium**	**Na** & **K** are bulk elements exerting fundamental functions in relaying information and keeping homeostasis in intra- and extra-cellular fluids. The most common anion composing Na and K salts is chlorine. Their role in the skin is the general one (without much relief) and the only local particularity consists in incorporation of sodium and potassium salts, as well as lactic acid, secreted by sweat glands, into the superficial stratum corneum. Sodium and potassium lactates contribute to the natural moisturizing factor, particularly responsible for the water holding capacity in these uppermost epidermal cell layers. In fact, in the course of skin thermoregulatory function, substantial quantities of body electrolytes are lost with sweat.	[171,172,173,174]
**Iron**	Numerous enzymatic reactions are **Fe**-dependent, e.g., cytochrome P 450, peroxidases, lipoxygenases, dioxygenases, or nitric oxide synthase. In this way, iron takes part in scavenging of various free radicals. This adds to the well-known functions of iron in oxygen transport and storage. Skin relevant procollagen-proline dioxygenase incorporates oxygen to the organic substrate in the presence of Fe^2+^.	[51,175]
**Zinc**	**Zn** is required for function of over 300 enzymes implicated in various fundamental biological processes like signal transduction, gene transcription, maintaining of DNA integrity, protein folding, etc. Together with copper, it acts as part of anti-radical detoxification enzymes, extracellular and cytoplasmic superoxide dismutases. MMPs, important for tissue remodeling are enzymes containing a zinc-finger motif. Keratinocyte mobility and proliferation are enhanced by Zn, and so are synthesis and reorganization of collagen fibers. Anti-inflammatory action of ZnO is due to lowering levels of iNOS and NO production.	[13,31,32,86,94,95,105,106,158,176]
**Copper**	**Cu** is essential for several fundamental processes due to its cofactor function in numerous enzymes, e.g., lysyl oxidase, superoxide dismutase 1, EC-superoxide dismutase, amine oxidase, cytochrome C oxidase, tyrosinase. Thus, copper is indispensable for angiogenesis, oxygen transport, energy production, antioxidant defence, iron metabolism, immunity, and pigmentation. Together with Zn and Mn, it stimulates expression of beta 1 integrins, increasing thus keratinocyte mobility and regenerative potential of the epidermis.	[86,93,152]
**Manganese**	**Mn** is a cofactor in the mitochondrial superoxide dismutase 2 and other metalloenzymes involved in antioxidant action, protein, and energy metabolism. It is also involved in biosynthesis of fibrillary collagen glycosaminoglycans, as a cofactor of galactosyl and glucosyl transferases. Up-regulation of keratinocyte integrins by Mn ions leads to an accelerated closure of epithelial wounds in vitro.	[86,92]
**Molybdenum**	**Mo** containing enzymes oxidize and detoxify purines and pyrimidines, catalyze the transformation of potentially noxious sulfites to innocuous sulfates and the conversion of hypoxanthine to uric acid. Mo–dependent enzymes and their co-factors have evolved to support coded life with its complex genetic schemes.	[177]
**Selenium**	**Se** is a component of numerous enzymes that catalyzes redox reactions, e.g., glutathione peroxidases, thioredoxin reductases and methionine-R-sulfoxide reductase. Phospholipid hydroperoxide glutathione peroxidase is particularly important for detoxification of cell membranes.	[52]
**Silicium**	**Si** salts used as a nutritional supplement are beneficial for bone formation and collagen 1 production. Their positive impact on dermal regeneration and anti-ageing effect have thus been suggested. A possibility of silicon involvement in connective tissue stabilization or formation has been inferred from the ease with which stable polyvalent silicon complexes can be formed with sugar-like molecules in an aqueous milieu.	[178,179]

## Figures and Tables

**Figure 1 ijms-23-06267-f001:**
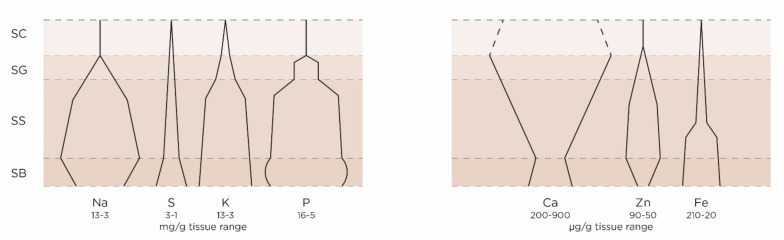
Distribution of some relevant minerals in human epidermis (relative quantities, gradients) mg/g range: Na, S, K, P; µg/g range: Ca, Zn, Fe, (Mg: quantitative data not found; Cu, below the method’s sensitivity limit). Based on data found in the literature. Due to the differences in methodological approaches, this “skyline” graph shows only a relative abundance of various elements, as they are distributed throughout various epidermal layers (SB = *stratum basale*; SS = *stratum spinosum*; SG = *stratum granulosum;* SC = *stratum corneum*). Most of the minerals are not detected or in very low quantities in the horny layer. No consensus has been reached concerning calcium; depending on the method of detection, some groups continue to see it in the SC, the others claim its complete absence at this location in case of persistence of the functional permeability barrier.

**Figure 2 ijms-23-06267-f002:**
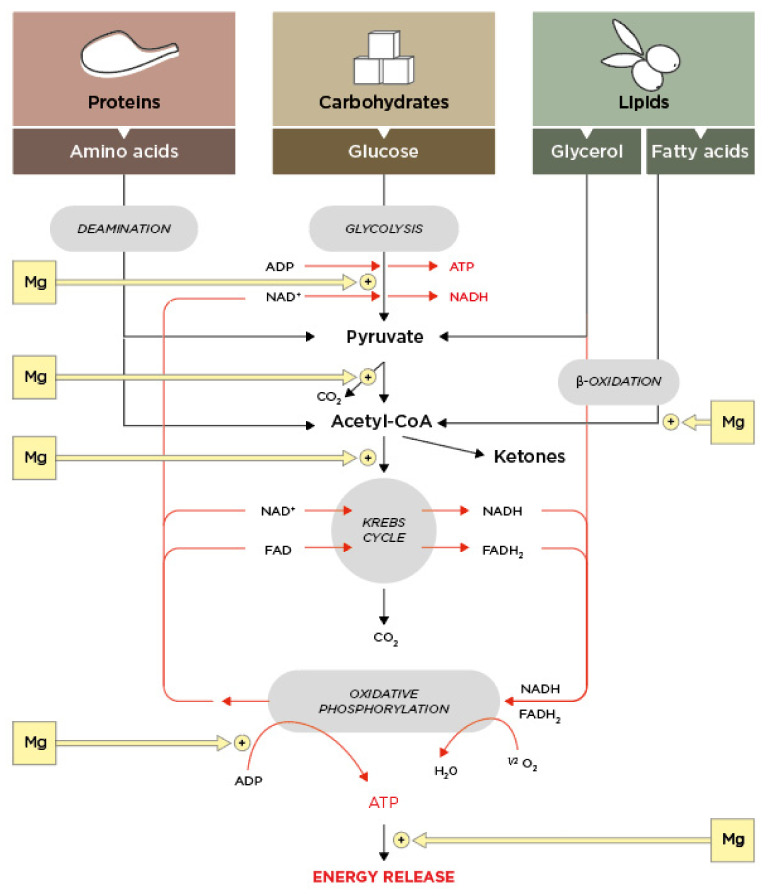
Implication of magnesium in cell bioenergetics.

**Table 1 ijms-23-06267-t001:** Mineral macroelements, oligoelements, and trace elements in human skin (relative quantity per whole body weight).

% Whole Body Weight	Chemical Composition
**Bulk organic elements** (95%)	**C, O, H, N**
**Macroelements** (<5%)	**Na, K, Mg, Ca, P, S, Cl**
**Oligoelements** (<0.1%)	**Cu, Zn, Se, Mn, Fe, Mo, Co**
**Oligoelements** (<0.01%)	**F, I, Sn**
Trace elements found on the skin surface	As, B, Cr, Ni, St, V; Al, Zr, Ag, Au, Hg, Si (silicates)

Essential elements are in bold characters. Trace elements found on skin surface are environmental contaminants.

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
