# Peer review of "Skin Minerals: Key Roles of Inorganic Elements in Skin Physiological Functions"

_ijms, 2022, doi:10.3390/ijms23116267_

Round 1
Reviewer 1 Report
The authors present a very interesting review on skin minerals and their functions. Overall the article is very well thought through and explained clearly. The review could be accepted in its present form.
Optional:
However, I believe the review could benefit if it included at least to some part discussing the importance of the minerals in common skin diseases such as atopic dermatitis / ezcema or psoriasis for example.
Moreover with increasing evidence of skin microbiome and their interactions with various elements of skin, it would be interesting to read, if the authors could briefly discuss how these minerals could benefit the resident microbes?
Author Response
We thank Reviewier 1 for her/his evaluation and pertinent suggestions.
We are pleased to learn that our review meets the high standards of the special issue on Current Topics in Trace Element and Mineral Research.
Indeed, skin pathology and skin microbiota constitute extremely interesting aspects, in which inorganic elements interact. We retain the Reviewer’s suggestion to develop upon in our future publications. We believe that the present paper, focused on the basics of skin physiology, will serve as a perfect launching pad for further developments.
Reviewer 2 Report
The review is well structured and the topic is certainly of interest to readers.
I suggest a minor revision, mainly focused on the following points:
- a large number of abbreviations can be found along the text; Authors should provide an initial nomenclature list for easier reading of the manuscript.
- the title is centered on "skin mineral". Based on the definition of "mineral", I don't know if there could be a better title to include all the references cited in the review, sometimes not strictly related to "minerals" as per definition
- some chapters (for example 3.1) describe effects resulting from physiological mechanisms that should be more clearly traced back to mineral skins
Author Response
The authors perfectly agree with the Reviewer 2 idea to add Abbreviations’ list in order to facilitate reading of the paper.
The list of principal abbreviations used in our paper is now introduced, at
the end of the manuscript before the references, in accordance with the IJMS rules.
Concerning the definition of mineral, and use of this word in our text, we agree that it may vary according to the centre of interest of an author (or that of a reader). According to Collin’s (https://www.collinsdictionary.com/dictionary/english/mineral), there are two main levels of comprehension of this word: 1) any of a class of naturally occurring solid inorganic substances with a characteristic crystalline form and a homogeneous chemical composition; 2) any inorganic matter. Indeed, we decided to play upon this semantic ambiguity to make the story (and the title) more attractive. By the way, the subtitle clearly defines the scope of our review.
A more ‘popular’ meaning of ‘mineral’ as a noun is: “An inorganic element, such as calcium, iron, potassium, sodium, or zinc, that is essential to the nutrition of humans, animals, and plants” (https://www.yourdictionary.com/mineral ). As our review is intended to address the fundamental roles of several inorganic elements in skin biology, we admit to preferentially focus on this latter definition. As explained from the very beginning of our text (paragraph 2.), mineral elements and compounds are usually useful for cells and tissues in their ionic state. We trust that this fact indicates clearly that we do not consider dry mineral matter being involved in most of the described physiological events. Just in case, we added a few words in paragraph 2 (lines 79-80), and it is valid for the entire text.
We hope that the aforementioned explanations satisfy concerns of Referee 2.
Reviewer 3 Report
The manuscript by Haftek et al., presents a very comprehensive and interesting review about the roles of inorganic elements on the physiology of skin. The information is also organized in a logical way and the mechanisms in which the different minerals exert their role on skin physiology are clearly described.
I have to point out the abstract. It begins in an appealing way, catching the reader´s attention. However, despite an excellent framing of the importance of minerals on skin physiology, it does not state clearly the goals of the work, neither the main conclusions, which many authors look for, when reading an abstract.
Other small details should be mentioned:
l.40 - "(Macro- and oligo- elements present in human skin)" - I do not understand why this sentence is here.
Table 1 - left collumn, last cell - the word skin is repeated.
Figure 1 - According to the Journal´s guidelines, the caption of figures must be placed after the image, not before.
lines 228 and 263 - the citations here are in the numerical and text form. please remove the text form.
Line 584 - The virus SARS-COV-9 is mentioned, but the reference is about a work with SARS-COV-2. Please correct.
Author Response
Many thanks for the positive evaluation of our review.
Although an Abstract is too frequently the only part of a paper to be read thoroughly, it remains a hard task to put in some 200 words all the message developed in an entire Review. We remain confident that the initial phrases of our Abstract, intended to captivate readers’ curiosity while summing up the central idea behind the story, will do the job. Indeed, the goal of this review is to draw attention to the inorganic factors in skin physiology and the conclusion is that mineral elements play several crucial roles therein.
The details to be corrected, kindly pointed out by the Reviewer, have been addressed adequately.